# Unrecognised COVID-19 deaths in central Europe: The importance of cause-of-death certification for the COVID-19 burden assessment

**Agnieszka Fihel** [1,2]*, **Anna Janicka**[1], **Andrea Buschner**[3], **Rūta Ustinavičienė**[4], **Aurelija Trakienė**[4]

**1** University of Warsaw, Warsaw, Poland, **2** Institut Convergences Migrations, Aubervilliers, France, **3** Bayerisches Landesamt für Statistik, Fürth, Germany, **4** Causes of Death Registry, Institute of Hygiene, Vilnius, Lithuania

\* a.fihel@uw.edu.pl

## Abstract

### Objective

In Central Europe, the increase in mortality during the COVID-19 pandemic exceeded the number of deaths registered due to coronavirus disease. Excess deaths reported to causes other than COVID-19 may have been due to unrecognised coronavirus disease, the interruptions in care in the overwhelmed health care facilities, or socioeconomic effects of the pandemic and lockdowns. Death certificates provide exhaustive medical information, allowing us to assess the extent of unrecognised COVID-19 deaths.

### Materials and methods

Data from 187,300 death certificates with a COVID-19 mention from Austria, Bavaria (Germany), Czechia, Lithuania and Poland, 2020–2021, was used. The two step analysis uses Cause of Death Association Indicators (CDAIs) and Contributing CDAIs to identify and measure the statistical strength of associations between COVID-19 and all other medical mentions.

### Results

15,700 deaths were reported with COVID-19 only as a contributing condition (comorbidity). In three cases out of four, a typical, statistically significant coronavirus complication or preexisting condition was registered as the underlying causes of death. In Austria, Bavaria, Czechia and Lithuania the scale of COVID-19 mortality would have been up to 18–27% higher had COVID-19 been coded as the underlying cause of death. Unrecognised coronavirus deaths were equivalent to the entire surplus of excess mortality beyond registered COVID-19 deaths in Austria and the Czech Republic, and its large proportion (25–31%) in Lithuania and Bavaria.

**Data Availability Statement:** The individual data can be purchased by everyone from the statistical offices of Austria, Bavaria, the Czech Republic,

Lithuania and Poland. The authors of the study can provide the data aggregated by sex, age and cause of death, on request and under certain conditions. Such an aggregation will make the full replication of the study possible. Don't hesitate to get in touch with the Centre of Migration Research, University of Warsaw: migration.cmr@uw.edu.pl, phone no. +48 22 55 46 770, or the Chair of the CMR Ethics Committee dr Sara Bojarczuk: s.bojarczuk@uw.edu.pl.

**Funding:** This publication was supported by the University of Warsaw under the Priority Research Area V of the "Excellence Initiative – Research University" programme. The funder had no role in study design, data collection and analysis, decision to publish, or preparation of the manuscript."

**Competing interests:** The authors have no relevant financial or non-financial interests to disclose.

## Conclusions

Death certificates with typical coronavirus complications or comorbidities as the underlying causes of death and contributing COVID-19 mentions were plausibly unrecognized coronavirus deaths.

## Introduction

As of March 2024, more than 7 million deaths were registered worldwide due to coronavirus disease 2019 (COVID-19), caused by the SARS-CoV-2 virus [1]. In many countries and world regions, including Central Europe, the so-called excess mortality in 2020–2021 considerably surpassed the number of deaths registered due to COVID-19 [2–5], although the exact magnitude of the excess mortality is uncertain as different estimation techniques can be used. There are several reasons why the mortality during the pandemics exceeded the number of deaths reported due to COVID-19. First of all, some deaths attributable to COVID-19 may not have been certified as such, for example due to lack of testing, atypical course of disease or cause-of-death misclassification [6, 7], the latter of which was suggested for Latvia and Poland [7, 8]. Another source of bias is deaths from other causes, such as acute cardiovascular conditions and neoplasms, which may have increased as the overwhelmed health care systems prioritised coronavirus patients [9–14], or decreased due to the elimination of risk factors such as air pollution, heavy traffic, or communicable diseases [5, 15]. Finally, mortality may have increased due to harmful behaviours typical of the socioeconomic instability experienced by some groups during the pandemics, lockdowns and economic slowdown, such as abuse of noxious substances, suicides and accidents [16–18]. This study refers to the first of the above plausible explanations for the excess mortality: the unrecognised COVID-19 deaths.

All developed countries applied the WHO specimen of a death certificate which includes several sections describing the medical conditions that contributed to death. The so-called Part 1 includes the underlying cause of death (UCoD), that is, "the condition initiating the train of morbid events directly leading to death" [19], which is further reported by national statistical offices and the WHO, and the complications of the UCoD, which form the logical and chronological chain of events leading to death. Part 2 reports all other diseases, injuries, risk factors or conditions, known as comorbidities, that may have aggravated the progression of the chain of events described in Part 1. WHO guidelines for certification and coding of COVID-19 favour its selection as the UCoD: a death due to coronavirus disease "may not be attributed to another disease and should be counted independently of preexisting conditions that are suspected of triggering a severe course of COVID-19" [20]. Whether confirmed by diagnostic tests or only suspected, the WHO strongly advocates that SARS-CoV-2 infection be assigned as the UCoD when consistent clinical symptoms suggest infection, "unless there is a clear alternative cause of death that cannot be related to COVID disease" (Ibidem). Still in line with the WHO recommendations, guidance in the US explicitly stipulated that pre-existing conditions that did not cause coronavirus disease "but can increase the risk of contracting a respiratory infection and death" should be reported as comorbidities [21]. However, these recommendations were not published until April 2020 and were not fully enforced until the following months of the pandemic [16]. This delay, like any change in the certification rules leading to important distortions in cause-of-death mortality, resulted in some coronavirus deaths being assigned to other UCoDs [22].

Studies that estimate the excess mortality during the pandemics beyond registered COVID-19 deaths do not distinguish between different factors underlying excess mortality during the

pandemic [3] and do not consider the unrecognised coronavirus deaths separately from other deaths. In countries where detailed information on the time of death is available, unrecognised coronavirus deaths can be estimated based on statistical correlations between time series of deaths due to COVID-19 and other natural causes [23]. In countries where no such information is available, the multiple cause-of-death (MCoD) approach is useful. This approach uses all information reported in death certificates: the UCoDs, their complications and comorbidities, thus allowing analysis of cases where COVID-19 was certified as a comorbidity, rather than the UCoD. The MCoD data have been so far used to identify the most frequent comorbidities and complications of COVID-19 [24–31], or to demonstrate the competing effect between the SARS-CoV-2 virus and chronic diseases as causes of death. In selected regions of Brazil, Italy, Spain and the USA, the reporting of cardiovascular diseases, chronic obstructive pulmonary disease, diabetes and neoplasms as the underlying causes of death (UCoD) decreased [32–36] during the pandemics, but the prevalence of these conditions as comorbidities in death certificates increased, as they are also pre-existing conditions that exacerbate the course of COVID infection.

The objective of the present study was to investigate the medical circumstances in which COVID-19 was mentioned as a comorbidity and not as the UCoD, and thus to estimate the number of COVID-related deaths that contributed to excess mortality during the pandemic, but were not registered as coronavirus fatalities. The analysis drew on the most recent data available for the general populations of four Central European countries and one German federal state. National guidelines for certification of medical causes of death, including COVID-19, followed the WHO guidelines in all these countries. However, the practices used by medical doctors to report comorbidities and complications, particularly the level of detail, tend to vary internationally. Therefore, the first step of the analysis aimed to investigate deaths due to COVID-19 as the UCoD in order to identify the most frequent complications and comorbidities of the coronavirus fatal course in Central Europe. The Cause of Death Association Indicator (CDAI) was used to measure the strength of associations between COVID-19 and other reported conditions at the general population level [37]. The second step of the analysis concerned deaths in which COVID-19 was reported as a contributing condition; based on previously identified, statisticall significant complications and comorbidities of coronavirus, the associations between contributing COVID-19 and the UCoDs were examined. To this end, an original indicator, the Contributing Cause of Death Association Indicator (CCDAI) was proposed to allow for unusual certification practices observed in Central Europe, where COVID-19 was reported, correctly or not, as a comorbidity.

## Materials and methods

### Data

This study examines population-level mortality in Austria (years 2020–2021), Czechia (2020–2021), Lithuania (2020–2021) Poland (2021), and the German state of Bavaria (2020–2021), further collectively referred to as countries. Individual death records were obtained from the national registration systems of the respective countries and the Bavarian State Office for Statistics. The available information included sex, age at death, year and all the medical mentions registered in death certificates. Apart from the UCoD, medical mentions included the consequences of the UCoD (listed in Part I of the death certificates) and comorbidities (Part II), jointly referred to as contributing mentions in this paper. In all the studied countries, an unlimited number of contributing mentions may be reported. A list of exhaustive and mutually exclusive cause-of-death categories used in the analysis is provided in S1 Table. Following

other research on pathologies in COVID-19 patients, the study excludes persons deceased at ages under 30, for whom mortality is often due to other pre-existing conditions.

## Analytical approach

The Cause of Death Association Indicator (CDAI) was used to identify significant associations between COVID-19 as the UCoD and contributing conditions (complications or comorbidities) and to compare the strength of associations for different conditions. The CDAI is the ratio of the observed frequency of a given complication or comorbidity $c$ reported together with the UCoD $u$, and the mean frequency of cause $c$ among all underlying causes of death:

$$CDAI_{u,c} = \frac{\sum_x \frac{d_{u,c,x}}{d_{u,x}} * \bar{d}_x}{\sum_x \frac{d_{c,x}}{d_x} * \bar{d}_x} * 100,$$

(1)

where $d$ stands for the number of deaths, subscript $u$ refers to a given UCoD (COVID-19), subscript $c$ refers to a given comorbidity or complication, subscript $x$ stands for age and $\bar{d}_x$ is the arithmetic average number of deaths at age $x$. A CDAI significantly higher than 100 indicates that complication or comorbidity $c$ is more common than expected in deaths due to UCoD $u$.

For deaths with COVID-19 as a comorbidity and non-COVID-19 UCoDs, the Contributing Cause of Death Association Indicator (CCDAI) was established, defined as the ratio of the observed frequency of a given underlying cause $u$ when comorbidity $c$ is present, and the mean frequency of cause $u$ among all comorbidities:

$$CCDAI_{c,\,u} = \frac{\sum_x \frac{d_{u,c,x}}{d_{c,x}} * \bar{d}_{c,x}}{\sum_x \frac{d_{u,x}}{d_x} * \bar{d}_{c,x}} * 100,$$

(2)

where $d$ stands for the number of deaths, subscript $u$ refers to a given UCoD, subscript $c$ refers to a given comorbidity (COVID-19), subscript $x$ stands for age and $\bar{d}_{c,x}$ is the arithmetic average number of deaths at age $x$ with comorbidity $c$. A CCDAI significantly higher than 100 indicates that underlying cause $u$ is more common than expected in deaths with comorbidity $c$.

To allow comparability across countries, years and genders, the average distributions of deaths by age, $\bar{d}_x$ and $\bar{d}_{c,x}$, were obtained by taking the arithmetic average of the number of deaths in all countries and years under analysis, taking into account both genders. For calculation of the 95% confidence intervals for CDAIs and CCDAIs, it was assumed that the observed number of deaths follows a Poisson distribution, with the variance approximated by the number of deaths. Both indicators were calculated with reference to all non-external deaths. Deaths assigned to external UCoDs (ICD10 codes from V01 to Y98) were excluded from the analysis because comorbidities have a fundamentally different impact on lethal processes in external and non-external deaths. If the underlying and contributing mentions certified were the same with an accuracy of four digits of the ICD10 code, the contributing mention was not taken into account.

## Results

### Certification of COVID-19

A total of 1,771 thousand deaths were registered as due to a non-external UCoD in the populations and years studied, of which 231,000 deaths had coronavirus listed as the UCoD and 15,700 included coronavirus only as a comorbidity (Table 1). In each country, over 98% of

**Table 1. Number of deaths from non-external UCoDs at age 30 and over and average number of non-UCoD mentions in death certificates, by country and year.**

| Country Year | Number of deaths | | | | Average number of non-UCoD mentions | | | |
| --- | --- | --- | --- | --- | --- | --- | --- | --- |
| | Total | COVID-19[a] as the UCoD | COVID-19[a] as contributing[b] | COVID-19[a] as the UCoD or contributing[b] | All deaths | Deaths with COVID-19[a] as the UCoD | | |
| | | | | | | Age 30+ | Age 30–59 | Age 60+ |
| **Total** | 1,286,001 | 171,588 | 15,702 | 187,290 | 2.80 | 3.85 | 3.93 | 3.88 |
| *2020* | *385,786* | *26,812* | *6,562* | *33,374* | *3.43* | *4.03* | *4.03* | *4.07* |
| *2021* | *900,215* | *144,776* | *9,140* | *153,916* | *2.53* | *2.71* | *2.60* | *2.74* |
| **Austria** | 172,716 | 14,333 | 2,740 | 17,073 | 3.41 | 4.12 | 4.18 | 4.20 |
| *2020* | *86,138* | *6,487* | *1,322* | *7,809* | *3.37* | *4.01* | *4.07* | *4.08* |
| *2021* | *86,578* | *7,846* | *1,418* | *9,264* | *3.44* | *4.22* | *4.22* | *4.30* |
| **Bavaria** | 276,034 | 19,550 | 3,801 | 23,351 | 4.30 | 4.56 | 5.00 | 4.54 |
| *2020* | *135,779* | *7,531* | *1,632* | *9,163* | *4.11* | *4.38* | *4.70* | *4.37* |
| *2021* | *140,255* | *12,019* | *2,169* | *14,188* | *4.49* | *4.68* | *5.09* | *4.65* |
| **Czechia** | 257,000 | 35,951 | 6,444 | 42,395 | 2.86 | 3.72 | 3.64 | 3.75 |
| *2020* | *123,054* | *10,530* | *2,610* | *13,140* | *2.80* | *3.67* | *3.47* | *3.70* |
| *2021* | *133,946* | *25,421* | *3,834* | *29,255* | *2.91* | *3.74* | *3.68* | *3.77* |
| **Lithuania** | 85,882 | 9,257 | 2,483 | 11,740 | 3.40 | 5.03 | 4.91 | 5.17 |
| *2020* | *40,815* | *2,264* | *998* | *3,262* | *3.17* | *4.64* | *4.50* | *4.76* |
| *2021* | *45,067* | *6,993* | *1,485* | *8,478* | *3.61* | *5.16* | *5.03* | *5.30* |
| **Poland[c]** | | | | | | | | |
| *(2020)* | *(485,259)* | *(41,451)* | *n.a.* | *n.a.* | *n.a.* | *n.a.* | *n.a.* | *n.a.* |
| *2021* | *494,369* | *92,497* | *234* | *92,731* | *1.62* | *1.86* | *1.84* | *1.86* |

[a]ICD-10 codes for COVID-19 from U07.1 to U10.9;

[b]Contributing mentions include the consequences of the UCoD (listed in Part I of the death certificates) and comorbidities (Part II), but exclude records with COVID-19 as the UCoD;

[c]Data for Poland in 2020 were not available and were not included in the analysis.

Source: own calculations based on data obtained from population register systems of Austria, Bavaria, Czechia, Lithuania and Poland.

deaths due to a coronavirus UCoD were assigned to one of the ICD10 codes that require a positive test for SARS-CoV-2: U07.1 and U08-U10. For all non-external deaths, the average number of contributing mentions ranged from 1.62 in Poland to 4.30 in Bavaria, but for deaths due to COVID-19 the respective figures were higher everywhere, ranging from 1.86 in Poland to 5.03 in Lithuania. Fatalities due to coronavirus UCoDs were mostly registered in hospitals or nursing homes, accounting for 78% of COVID-19 deaths in Austria, 95% in Lithuania, and 91% in Poland (no information available for Bavaria and Czechia).

## COVID-19 reported as the underlying cause of death

In deaths where COVID-19 was the UCoD, ten groups of comorbidities were found to be significantly associated with coronavirus disease (CDAI CI lower bound above 100, Fig 1 and S2 Table). Neoplasms and obesity, and in Austria and Bavaria, disorders involving the immune mechanism were reported more than twice as often for COVID-19 deaths as for all non-external deaths (CDAIs of over 200). Other comorbidities, such as infectious, Alzheimer, cardiovascular and chronic lower respiratory diseases, were also significantly associated with coronavirus in all the study countries except for Poland. As regards complications, ten groups of symptoms were found to be significantly associated with coronavirus as the UCoD, the most important (CDAIs over 200) and prevalent in most countries being influenza,

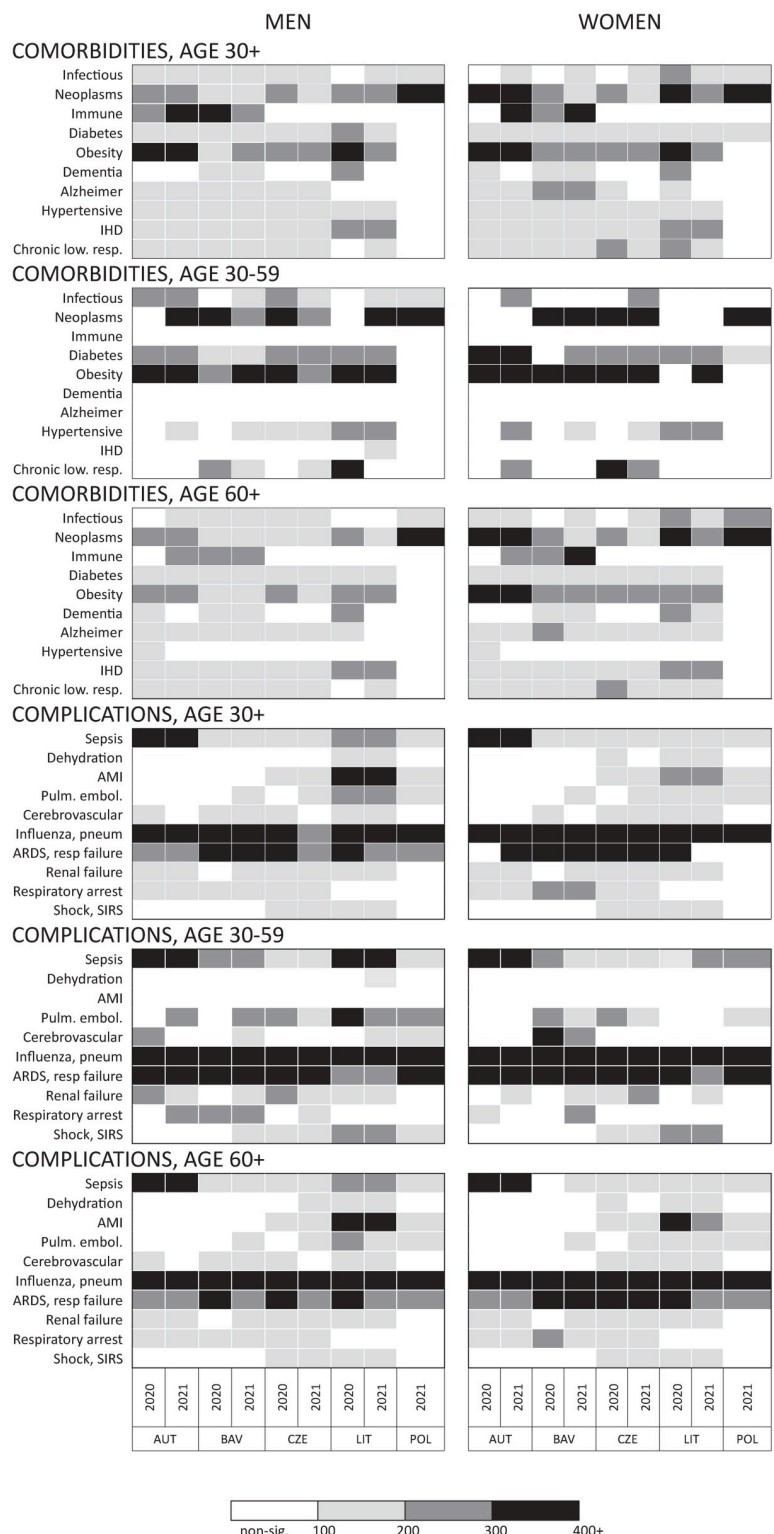

**Fig 1. Cause of Death Association Indicators[a] for selected comorbidities[b] and complications[c] in deaths due to COVID-19 (UCoD) at age 30 and over as compared to all non-external deaths, by sex, age, country and year.**
[a]CDAI values for which the lower bound of the 95% CI is below 100 and/or those corresponding to fewer than 50 cases are shown in white; [b]Well defined infectious and parasitic diseases (ICD-10 codes: A00-B99 excl. A40-A41, A48-A49, B34, B97-B99), malignant neoplasms (C00-C97), certain disorders involving immune mechanism (D80-D89), diabetes

mellitus (E10-E14), obesity (E66), dementias (F00-F09), Alzheimer disease (G30-G32), hypertensive diseases (I10-I15), Ischaemic Heart Disease (I20, I22-I25), chronic lower respiratory diseases (J40-J47); [c]Sepsis and other bacterial infections and viral agents (A40, A41, A48-A49, B34, B97-B99), dehydration (E86-E87), Acute Myocardial Infarction (I21), pulmonary embolism (I26), cerebrovascular diseases (I60-I69), influenza, pneumonia (J09-J18), Adult Respiratory Distress Syndrome and respiratory failure (J80-J81, J96), renal failure (N17-N19), respiratory arrest (R09), shock and Systemic Inflammatory Response Syndrome (R57, R65). Interpretation: a CDAI of 100 to 200 for a well-defined infectious disease in men aged 30 and over means that the frequency of that comorbidity in COVID-19 deaths is significant and up to twice as high as in all non-external deaths. Source: own calculations based on data obtained from population register systems of Austria, Bavaria, Czechia, Lithuania and Poland.

pneumonia, Adult Respiratory Distress Syndrome and respiratory failure, and, in younger persons, sepsis and pulmonary embolism.

## COVID-19 reported as a comorbidity

Deaths with COVID-19 mentioned only as a comorbidity constituted 8% of deaths with an underlying or contributing COVID-19 mention. Contributing coronavirus mentions constituted 21% of all COVID-19 mentions in Lithuania, 16% in Austria and Bavaria, 15% in Czechia and 0.3% in Poland. In Bavaria, Czechia, Lithuania and Poland, the majority of contributing coronavirus mentions had been confirmed by a SARS-CoV-2 test (ICD10 code U07.1) (Table 2); in Austria, only 48% of contributing coronavirus mentions had been confirmed by tests, whereas the remaining cases were due to either non-confirmed disease (U07.2), or an earlier confirmed or probable COVID-19 episode (U08-U10).

A descriptive analysis of deaths with a contributing COVID-19 mention shows that almost one in five deaths (19%) was assigned to a UCoD representing one of the typical coronavirus complications identified in the analysis presented in the previous section (listed in a footnote to Fig 1): cerebral infarction, Acute Myocardial Infarction, other cerebrovascular diseases and renal failure. A further 54% of deaths with a contributing COVID-19 mention were due to comorbidities that exacerbate the course of COVID infection and that were significantly associated with COVID-19 (listed in Fig 1): Ischaemic Heart Disease, neoplasms, diabetes mellitus, obesity, dementias, Alzheimer disease and chronic lower respiratory diseases.

The Contributing CDAIs indicate that the occurrence of well-defined infectious diseases, obesity, diabetes, disorders due to psychoactive substance use, and diseases of the circulatory system, particularly cerebrovascular diseases, as the UCoD was significantly more frequent in deaths with contributing COVID-19 than in all non-external deaths (Fig 2). Furthermore, COVID-19 was more frequently reported as a comorbidity in deaths due to complications and

**Table 2. COVID-19 contributing mentions in deaths at age 30 and over by country, counts and percentages.**

| Contributing mentions | Total | Austria | Bavaria | Czechia | Lithuania | Poland |
|---|---|---|---|---|---|---|
| | | | | N | | |
| **All COVID-19 mentions (U07.1-U10.9)** | 15,702 | 2,740 | 3,801 | 6,444 | 2,483 | 234 |
| | | | | % | | |
| **All COVID-19 mentions (U07.1-U10.9)** | 100.0 | 100.0 | 100.0 | 100.0 | 100.0 | 100.0 |
| *Therein* | | | | | | |
| **Virus identified (U07.1)** | 82.2 | 47.6 | 76.6 | 94.0 | 99.3 | 72.2 |
| **Virus not identified (U07.2)** | 9.6 | 26.0 | 10.9 | 5.1 | 0.7 | 14.1 |
| **Complications of COVID-19 (U08-U10)** | 8.2 | 26.4 | 12.5 | 0.9 | 0.0 | 13.7 |

Source: own calculations based on data obtained from population register systems of Austria, Bavaria, Czechia, Lithuania and Poland.

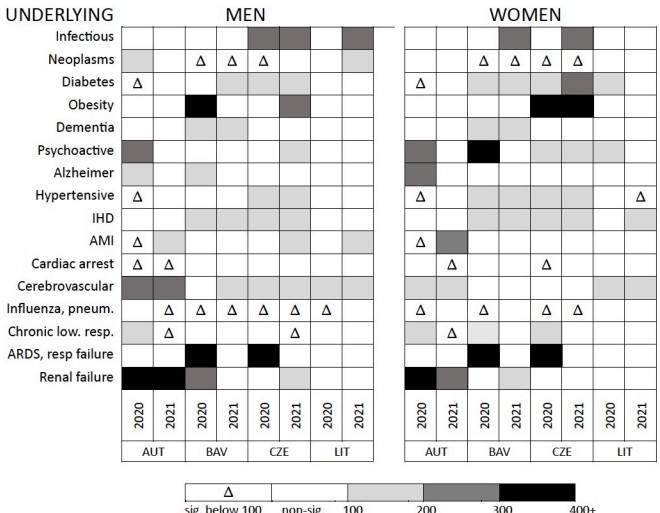

**Fig 2. Contributing Cause of Death Association Indicators[a] for selected underlying causes[b] in deaths with COVID-19 as comorbidity, as compared to all non-external deaths by sex, country[c] and year.** [a]The presented CDAIs are significantly below 100 (the upper bound of the 95% CI below 100) or significantly above 100 (the lower bound of the 95% CI above 100) and correspond to at least 50 cases. Non-significant CDAIs are shown in white. [b]Disorders due to psychoactive substance use (F10-F19), cardiac arrest and heart failure (I46, I50), for other ICD10 codes see notes to Fig 1; [c]Poland excluded due to low number of death counts with COVID-19 as a comorbidity. Interpretation: A CCDAI of 100 to 200 for a well-defined infectious disease means that the frequency of that disease is significant, being up to twice as high as in other underlying causes of death where COVID-19 was a comorbidity. Source: own calculations based on data obtained from population register systems of Austria, Bavaria, Czechia and Lithuania.

ill-defined mechanisms whose application as UCoDs should always be avoided: Adult Respiratory Distress Syndrome, respiratory failure and renal failure. On the other hand, COVID-19 contributing mentions were less frequently certified in deaths due to neoplasms, influenza and pneumonia and practically did not occur in deaths due to cardiac arrest, heart failure or ill-defined causes (ICD10 codes starting with 'R').

When the magnitude of deaths with COVID-19 as a contributing mention is related to existing estimates of excess mortality (Table 3), it may be seen that in Austria, where excess deaths in 2020–2021 are estimated to have numbered between 13,000 and 18,300, the addition of contributing mentions would increase the COVID-19 burden from 14,400 to 17,200 and allows almost all excess deaths to be accounted for. This is also the case for Czechia, where the estimates range from 34,100 to 49,100, and adding COVID-19 contributing mentions would increase the COVID-19 burden from 36,000 to 42,500. However, for Lithuania only the lowest of all the estimates, which vary between 11,300 and 20,000, corresponds to the number of COVID-19 mentions anywhere on the death certificate (11,859). The only estimate of excess mortality available for the federal state of Bavaria, by Wang et al. [4], is 34,600, which greatly surpasses the number of underlying and contributing mentions combined (23,384). For Poland, surplus mortality in 2021 was estimated by the WHO at 100,100 (mean estimate), ranging from 94,500 (low estimate) to 105,700 (high estimate), which is slightly above the number of all COVID-19 mentions registered in that year. Although for the studied countries the estimates of excess mortality surpassed the number of deaths registered with COVID-19 as the UCoD, in all of them except Poland taking into account also deaths with contributing COVID-19 could explain a large part of the difference.

**Table 3. Number of deaths due to COVID-19 deaths (UCoD or contributing) and excess mortality estimates (due to COVID-19 and other causes), all ages, by country and year.**

| Country / Year | COVID-19[a] as the UCoD | COVID-19[a] as contributing[b] | Karlinsky, Kobak [5] updated by Levitt et al. [3] | Economist updated by Levitt et al. [3] | Wang et al. [4] | WHO [38] (mean estimate) | Levitt et al. [3] |
|---|---|---|---|---|---|---|---|
| **Austria** | 14,354 | 2,845 | 15,261 | 16,877 | 18,300 | 15,974 | 13,007 |
| *2020* | *6,491* | *1,323* | | | | *8,087* | |
| *2021* | *7,863* | *1,522* | | | | *7,886* | |
| **Bavaria** | 19,573 | 3,811 | n.a. | n.a. | 34,600 | n.a. | n.a. |
| *2020* | *7,537* | *1,637* | | | | | |
| *2021* | *12,036* | *2,174* | | | | | |
| **Czechia** | 35,994 | 6,459 | 35,000 | 43,942 | 49,100 | 43,248 | 34,079 |
| *2020* | *10,539* | *2,612* | | | | *16,600* | |
| *2021* | *25,455* | *3,847* | | | | *26,648* | |
| **Lithuania** | 9,284 | 2,575 | 16,008 | 17,396 | 20,000 | 17,447 | 11,283 |
| *2020* | *2,266* | *1,023* | | | | *6,154* | |
| *2021* | *7,018* | *1,552* | | | | *11,293* | |
| **Poland** | 133,948 | | 157,247 | 171,806 | 214,000 | 163,144 | 149,722 |
| *2020* | *41,451* | *n.a.* | | | | *63,032* | |
| *2021* | *92,780* | *234* | | | | *100,112* | |

[a]ICD-10 codes for COVID-19 from U07.1 to U10.9;

[b]Excluding records with COVID-19 as the UCoD.

Source: own calculations based on data obtained from population register systems of Austria, Bavaria, Czechia, Lithuania and Poland.

Deaths with contributing COVID-19 constituted 25% of the excess, non-coronavirus deaths in Bavaria, 31% in Lithuania (the WHO mean estimate) and 100% in Austria and the Czech Republic. Had all coronavirus contributing conditions been certified as the UCoD, registered COVID-19 mortality would have been higher by 19% in Austria and Bavaria, 18% in Czechia and 27% in Lithuania. In a more conservative scenario, if only deaths due to typical COVID-19 complications (as listed in Fig 1) were considered as unrecognised COVID deaths, registered coronavirus mortality would be higher by 5% in Austria, 4% in Czechia and 7% in Lithuania.

## Discussion

In the studied Central European countries, coronavirus deaths were certified predominantly as the UCoDs, confirmed by a positive COVID-19 test result, and reported with more cause-of-death mentions on average than other non-external deaths. Even in Poland, where the quality of cause-of-death data is generally relatively poor [39], more detailed and conclusive medical information was provided for coronavirus deaths than in certificates reporting other underlying causes of death. The fact that most of the COVID-19 deaths took place in hospitals or nursing homes may have contributed to the relatively good quality of medical documentation.

The study identifies the most important complications aggravating the progression of coronavirus disease and leading to death, such as sepsis, pulmonary embolism, influenza, pneumonia, ARDS and respiratory failure, which is in line with previous prospective cohort studies [40–47] and Multiple Cause-of-Death studies for Brazil, Italy, and the United States [24, 28–31, 48]. The present study, however, enhances the existing research by using CDAIs to demonstrate that these complications are not only the most common in COVID-19 deaths, but also

significantly more frequent than in other non-external deaths. While COVID-19 complications show a high degree of consistency across age, sex and study country, some differences between age groups can be observed in the frequency of reported comorbidities. The strongest associations with COVID-19 were identified for cardiovascular diseases, neoplasms and obesity.

Most recent studies aim to assess the number of unrecognised COVID-19 deaths [23], but the COVID-19 mentions reported as a comorbidity have never been examined for this purpose. Our study found that 8% of death certificates citing COVID-19 reported it only as a comorbidity, more frequently in Austria, Bavaria, Czechia and Lithuania than in Poland. The real mortality burden of COVID-19 in the first four countries would have been from 4–7% (if only deaths due to complications are considered as miscertified) to 18–27% (if all deaths with coronavirus mentions are considered as such) higher had COVID-19 been coded as the UCoD, as was done in Poland. Coronavirus contributing mentions accounted for the entire observed difference between registered COVID-19 mortality and estimated excess mortality in Austria and Czechia, but not in Lithuania and Poland, which are often mentioned as having important mortality surpluses [2, 3, 9]. In these countries, the elevated mortality in the pandemic period should be attributed to other causes of death or to unrecognised COVID-19 cases, but the different diagnostic testing policies implemented by Lithuania and Poland make it difficult to draw broad comparisons [49]. Nevertheless, a report from the Ministry of Health of Poland [50] suggests that the number of unrecognised COVID-19 cases in that country was considerable in 2020, the year for which data on multiple causes of death are not available. According to this analysis, 67,000 excess deaths were reported in 2020, of which 43% were due to COVID-19 as the UCoD and 27% due to other UCoDs in coronavirus patients. To recognise the overriding public health interest at the time, in the following year the Ministry decided both to tighten the cause-of-death coding rules, giving priority to COVID-19 as the UCoD, and to digitise the MCoD data, which, with one exception [39], had never been available before.

The MCoD approach is usually applied to general national populations and as such it provides full coverage and comprehensive information on mortality [37]. However, the results and their international comparability depend on the certification practices used in each country. In this study, the number of medical mentions reported together with COVID-19 varied considerably between countries, and so did the numbers of statistically significant comorbidities associated with COVID-19. Another evident limitation of the MCoD approach is that it relates cause-specific mortality to overall mortality rather than to the population at risk. This framework does not allow for causal inference on COVID-19 infection mechanisms. Instead, it enhances current clinical and epidemiological research by incorporating contributing mentions, which, as the study is original in showing, are prevalent even in the case of COVID-19, despite clear guidelines of the WHO.

Building on the most recent suggestions that complex analytical methods should be applied to COVID-related mortality [51], this investigation uses robust statistical metrics of the Cause of Death Association Indicator and proposes an additional form of this metric, the Contributing Cause of Death Association Indicator, which takes into account situations where coronavirus is reported only as a contributing factor. This original measure can be particularly useful in other MCoD studies that investigate causes of death in situations where certification practices change or diagnoses remain ambiguous, resulting in a medical condition being reported as both the UCoD and a contributing condition. As both metrics are standardized by age, the effect of the age pattern of mortality in the reference population is neutralized.

For deaths due to typical adverse effects of COVID-19 or ill-defined conditions, and with coronavirus reported as a contributing condition, the choice of a non-coronavirus UCoD on

the death certificate is highly questionable. Similarly, certification practices raise concerns when typical COVID-19 comorbidities were selected as the UCoDs in the presence of coronavirus; this particularly applies to the comorbidities for which strong associations were revealed by the CCDAIs. It is therefore reasonable to conclude that according to WHO guidelines, an important proportion of these deaths in Central Europe could also have been reported as having the COVID-19 underlying cause. However, this conclusion cannot be directly extrapolated to other countries and regions of the world, as certification practices used be medical doctors vary internationally.

During the COVID-19 pandemics, mortality increased far beyond the reported deaths from coronavirus disease, and several factors were responsible for this. Many studies suggest that health care systems were either overburdened or entirely reserved for people diagnosed with coronavirus [9–14], which contributed to postponement of treatment of chronic diseases or neglect of acute conditions, especially cardiovascular symptoms [52]. Another strand of research points to the long-term health consequences of COVID-related closure restrictions and the economic downturn: stress and mental health problems, the misuse of harmful substances, and related self-harm and accidents [16–18]. While the importance of these factors can be approximated by studying mortality from causes other than COVID-19, the scale of unrecognised coronavirus cases has rarely been explored due to lack of data and methods. The most recent study [23] showing that peaks in COVID-19 deaths were preceded at short time intervals by peaks of all-cause mortality in some regions of the United States is a novelty in this respect. This study shows that despite the complexity of the pathophysiological process leading to death in the presence of SARS-CoV-2, the use of MCoD data and metrics allows us to overcome some of the ambiguity of certification practices and to assess the overall burden of COVID-19. In the Central European countries studied, taking into account deaths with contributing COVID-19 explained a large part of (in Bavaria and Lithuania) or all (in Austria and the Czech Republic) of the difference between the total excess mortality and deaths registered due to COVID-19.

## Supporting information

**S1 Table. COVID-19 and contributing conditions used in the analysis.**
(DOCX)

**S2 Table. Cause of Death Association Indicators for selected comorbidities and complications in deaths due to COVID-19 (UCoD) at age 30 and over as compared to all non-external deaths, by sex, age, country and year.**
(XLSX)

## Author Contributions

**Conceptualization:** Agnieszka Fihel, Anna Janicka.

**Data curation:** Agnieszka Fihel, Anna Janicka, Andrea Buschner, Rūta Ustinavičienė, Aurelija Trakienė.

**Funding acquisition:** Agnieszka Fihel.

**Investigation:** Agnieszka Fihel, Anna Janicka, Andrea Buschner, Rūta Ustinavičienė, Aurelija Trakienė.

**Methodology:** Agnieszka Fihel, Anna Janicka.

**Software:** Anna Janicka.

**Supervision:** Agnieszka Fihel.

**Visualization:** Agnieszka Fihel.

**Writing – original draft:** Agnieszka Fihel.

**Writing – review & editing:** Agnieszka Fihel, Anna Janicka, Andrea Buschner, Rūta Ustina-
vičienė, Aurelija Trakienė.

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
