## [Decision Letter · Decision Letter 0]

7 May 2024

PONE-D-24-12758Unrecognised COVID-19 Deaths in Central Europe: the Importance of Cause-of-Death Certification for the COVID-19 Burden AssessmentPLOS ONE

Dear Dr. Fihel,

Thank you for submitting your manuscript to PLOS ONE. After careful consideration, we feel that it has merit but does not fully meet PLOS ONE’s publication criteria as it currently stands. Therefore, we invite you to submit a revised version of the manuscript that addresses the points raised during the review process.

We look forward to receiving your revised manuscript.

Kind regards,

Dickens Otieno Onyango

Academic Editor

PLOS ONE

Journal Requirements:

   "This publication was supported by the University of Warsaw under the Priority Research Area V of the "Excellence Initiative – Research University" programme."

4. In this instance it seems there may be acceptable restrictions in place that prevent the public sharing of your minimal data. However, in line with our goal of ensuring long-term data availability to all interested researchers, PLOS’ Data Policy states that authors cannot be the sole named individuals responsible for ensuring data access (http://journals.plos.org/plosone/s/data-availability#loc-acceptable-data-sharing-methods).

Reviewers' comments:

Reviewer's Responses to Questions

**Comments to the Author**

1. Is the manuscript technically sound, and do the data support the conclusions?

Reviewer #1: Yes

Reviewer #2: Yes

2. Has the statistical analysis been performed appropriately and rigorously? 

Reviewer #1: Yes

Reviewer #2: Yes

3. Have the authors made all data underlying the findings in their manuscript fully available?

Reviewer #1: Yes

Reviewer #2: Yes

4. Is the manuscript presented in an intelligible fashion and written in standard English?

Reviewer #1: Yes

Reviewer #2: Yes

5. Review Comments to the Author

Reviewer #1: Thank you for this well-written and highly relevant article. I only have minor comments:

In the abstract, first sentence methods: I would add '(Germany)' behind Bavaria, also the sentence is missing a verb, e.g., 'Data from ... was used.'

Introduction: I would not consider traffic accidents as the risk factor but for example heavy traffic or something alike

Analytical approach: I did not quite get '...and the frequency expected if causes u and c were independent' but it might just be me, however if you do have any space left than just a bit more info would be great

For '... average distributions of deaths by age, ® and ®,, were calculated for the studied populations' I did not quite understand how population/death structure was taken into account (again may be me) [everything else was good to follow and nicely described]

Results: Tab1. - as Polands 2020 data are not considered (in the total) I would put them in brackets or grey, also you do state in the data section what 'contributing' means: every mentioning (part I or II death certifcate) of Cov19 except as UCoD, but I would put it again below the table (makes it easier for the reader to follow); also it could be an option to include a joint category of any mentioning of Cov19 on death certificate, which would sum up column 3 and 4, as you are refering to this in the first paragraph below fig. 1 (took me some calculating to get there) (similarily this could be done for Tab. 3)

also I do undstand Polands low value of 3‰ but would probably stick with % (again to not confuse the reader)

I would not mind a literature reference for the 'typical coronavirus complications'

'cerebral infarction, Acute Myocardial Infarction, other cerebrovascular diseases' - I would stick with lower case (in any case consistent would be good)

'was estimated by the WHO at between 94,500 and 105,700' I suppose this refers to the different WHO scenarios (as it is not visible in the table), you should probably add that info

Discussion: in the third paragraph you are refering to Polands excess deaths in 2020, whereas you are not looking at 2020 data beforehand, maybe stick with 2021 here as well? also I was not quite sure where those 67,000 excess deaths come from as they are not displayed in Tab. 3 (though it might be Ref. 50)

I personally, would have expected at least a little elaborating on other sources of excess mortality (delayed health care access etc.) and even reductions in mortality (e.g. less traffic accidents) in the discussion, you briefly mention those in the beginning but never get back to this and it should at least be acknowledged that other things have happend during pandemic times and not all excess can or should (?) be diretly attributed to COVID-19

Reviewer #2: Good Manuscript, highlights the role of death certification in evaluating excess deaths, in this circumstance, occurring during COVID, and highlights areas where proper medical certification of death, would have identified these mortalities. In addition, this paper highlights the burden of COVID mortality, among persons with co-morbidities, particularly obesity, and cancer diagnosis. It is particularly useful in the assessment of death occurring in these populations, and particularly, development of hypotheses, on how to protect these vulnerable persons. It also addresses aspects of COVID, that may address potential areas of misinformation, eg vaccine hesitancy.

6. PLOS authors have the option to publish the peer review history of their article (what does this mean?). If published, this will include your full peer review and any attached files.

Reviewer #1: No

Reviewer #2: **Yes: **Edwin Walong

---

## [Author Response · Author response to Decision Letter 0]

6 Jun 2024

To the Editor and the Reviewers of manuscript ‘Unrecognised COVID-19 Deaths in Central Europe: the Importance of Cause-of-Death Certification for the COVID-19 Burden Assessment’ (submission no. PONE-D-24-12758)

Dear Editor, Dear Reviewers,

We would like to express our sincere thanks for considering the paper for publication and giving us the opportunity to revise it. The Reviewers devoted considerable effort to reading our paper, and their comments were relevant and valuable in improving the manuscript.

In response to the Editor’s and the Reviewers’ comments and suggestions, we have made several changes to the manuscript, each of which is explicitly addressed in our responses. We have modified our manuscript in a change-tracking mode for easy reference during the review process. In the following sections of this letter, we provide point-by-point responses (in italics) to the concerns raised by the Editor and Reviewers.

We firmly believe that the manuscript has been significantly improved by the invaluable input of the Editor and Reviewers. We would like to thank you again for the time you have taken to review this manuscript.

Kind regards,

The manuscript authors

Journal Requirements:

Response: We ensured that the manuscript complies with PLOS ONE's style requirements.

Response: This mention has been removed accordingly.

 "This publication was supported by the University of Warsaw under the Priority Research Area V of the "Excellence Initiative – Research University" programme."

Response: We have added the recommended statement that the funder had no role in the study in the separate cover letter.

4. In this instance it seems there may be acceptable restrictions in place that prevent the public sharing of your minimal data. However, in line with our goal of ensuring long-term data availability to all interested researchers, PLOS’ Data Policy states that authors cannot be the sole named individuals responsible for ensuring data access (http://journals.plos.org/plosone/s/data-availability#loc-acceptable-data-sharing-methods).

Response: We have provided non-author contact details: telephone and email to the secretary of our institute, and email to the chair of the ethics committee of our institute. The data will be stored on the internal drive of the university in accordance with the data confidentiality policy.

Response: This has been corrected accordingly.

Response: The references have been double-checked. One reference was deleted because it appeared twice in the list of references. Two additional references have been added:

Bugger H, Gollmer J, Pregartner G, Wünsch G, Berghold A, Zirlik A, et al. Complications and mortality of cardiovascular emergency admissions during COVID-19 associated restrictive measures. Schäfer A, editor. PLoS ONE. 2020 Sep 24;15(9):e0239801.

WHO. Estimates of Excess Mortality Associated With COVID-19 Pandemic (as of 5 April 2023). Geneva: WHO; 2023.

7. If applicable, we recommend that you deposit your laboratory protocols in protocols.io to enhance the reproducibility of your results. Protocols.io assigns your protocol its own identifier (DOI) so that it can be cited independently in the future.

Response: Following this recommendation, we have published the protocol describing our method of analysis on protocols.io. In this letter, we do not provide the doi for reasons of confidentiality (double blind review). However, if our paper is published, we will link the paper to the protocol.

Reviewer #1: 

Thank you for this well-written and highly relevant article. I only have minor comments:

1. In the abstract, first sentence methods: I would add '(Germany)' behind Bavaria, also the sentence is missing a verb, e.g., 'Data from ... was used.'

Response: Thank you, both issues have been corrected accordingly.

2. Introduction: I would not consider traffic accidents as the risk factor but for example heavy traffic or something alike

Response: Yes, this was a mental shortcut. We corrected into ‘heavy traffic’.

3. Analytical approach: I did not quite get '...and the frequency expected if causes u and c were independent' but it might just be me, however if you do have any space left than just a bit more info would be great

Response: Again, this was a mental shortcut. The term ‘independent’ was used in a statistical sense, meaning that the co-occurrence of comorbidity c and underlying cause u is not significantly different from the co-occurrence of comorbidity c and other underlying causes. The authors who proposed the Cause-of-Death-Association Indicator originally argued that this measure is used to ‘identify unexpectedly frequent associations’ between causes of death (Désesquelles et al. 2010, p. 773) and that ‘many associations are indeed more frequent than would be predicted by a random occurrence of the diseases’ (Ibidem, p. 786). 

We modified our manuscript in the following way :

The sentence: ‘The CDAI is the ratio of the observed frequency of a given complication or comorbidity c reported together with the UCoD u, and the frequency expected if causes u and c were independent’ (page 8, lines 170-173)

was replaced by the sentence: ‘The CDAI is the ratio of the observed frequency of a given complication or comorbidity c reported together with the UCoD u, and the mean frequency of cause c among all underlying causes of death’

The sentence: ‘For deaths with COVID-19 as a comorbidity and non-COVID-19 UCoDs, the Contributing Cause of Death Association Indicator (CCDAI) was established, defined as the ratio of the observed frequency of a given underlying cause u when comorbidity c is present, and the frequency expected if cause u were independent of comorbidity c’ (page 8, lines 182-185)

was replaced by the sentence: ‘For deaths with COVID-19 as a comorbidity and non-COVID-19 UCoDs, the Contributing Cause of Death Association Indicator (CCDAI) was established, defined as the ratio of the observed frequency of a given underlying cause u when comorbidity c is present, and the mean frequency of cause u among all comorbidities’.

4. For '... average distributions of deaths by age, ® and ®,, were calculated for the studied populations' I did not quite understand how population/death structure was taken into account (again may be me) [everything else was good to follow and nicely described]

Response: These average distributions of deaths by age were used to remove the effect of the different age structure of deaths by underlying cause and gender, as recommended by Désesquelles et al. (2010). In our study, the d ®_x and d ®_(c,x), are the algebraic averages of death counts (by age x and cause c) in five populations and in both years under analysis (Poland 2020 was included in this standardisation step). This is equivalent to looking at the age distribution of the total number of deaths in all countries and both years, since in the CDAI formula, the average death term d ®_x appears both in the numerator and in the denominator. The inclusion of these averages in formulas no. (1) and (2) means that in effect we obtain for each country, year, gender and age a ‘standardised’ frequency of occurrence of a specific association of causes, which may be compared accross countries, years and genders.

To explain this step in our method, we modified our manuscript in the following way (page 9, lines 199-201): 

‘To allow comparability across countries, years and genders, the average distributions of deaths by age, d ®_x and d ®_(c,x), were obtained by taking the arithmetic average of the number of deaths in all countries and years under analysis, taking into account both genders. 

5. Results: Tab1. - as Polands 2020 data are not considered (in the total) I would put them in brackets or grey, also you do state in the data section what 'contributing' means: every mentioning (part I or II death certifcate) of Cov19 except as UCoD, but I would put it again below the table (makes it easier for the reader to follow); also it could be an option to include a joint category of any mentioning of Cov19 on death certificate, which would sum up column 3 and 4, as you are refering to this in the first paragraph below fig. 1 (took me some calculating to get there) (similarily this could be done for Tab. 3). also I do undstand Polands low value of 3‰ but would probably stick with % (again to not confuse the reader)

Response: Thank you for all these suggestions. We have amended Table 1 accordingly: we have added the detailed note on what ‘contributing’ means, added another column presenting the totals of deaths with any COVID-19 mention, and replaced 3‰ with 0.3%.

6. I would not mind a literature reference for the 'typical coronavirus complications'

Response: In the abstract and Results, the phrasing ‘typical coronavirus complications’ (‘COVID-19 reported as a comorbidity’ section) was used to refer to the most frequent medical conditions reported in the death certificates we analysed. In Discussion, when summarising this result, we provided (in the original manuscript) the references to other studies that identified the same COVID-19 complications (pages 15-16, lines 397-403):

‘The study identifies the most important complications aggravating the progression of coronavirus disease and leading to death, such as sepsis, pulmonary embolism, influenza, pneumonia, ARDS and respiratory failure, which is in line with previous prospective cohort studies [40–47] and Multiple Cause-of-Death studies for Brazil, Italy, and the United States [24,28–31,48].’

We hope that these references are sufficient.

7. 'cerebral infarction, Acute Myocardial Infarction, other cerebrovascular diseases' - I would stick with lower case (in any case consistent would be good)

Response: We acknowledge the lack of consistency but the capital letters are used according to the medical literature standards, where Acute Myocardial Infarction is often referred to as AMI, Ischaemic Heart Disease as IHD, Adult Respiratory Distress Syndrome as ARDS and Systemic Inflammatory Response Syndrome as SIRS. In contrast, general categories such as cerebral infarction are not referred to as CI.

8. 'was estimated by the WHO at between 94,500 and 105,700' I suppose this refers to the different WHO scenarios (as it is not visible in the table), you should probably add that info

Response: These figures refer to low and high estimates, whereas Table 3 only includes the mean estimate. In the revised manuscript, we have clarified this in the following way:

– The heading of Table 3 includes a note that the numbers refer to the mean estimate, and

– The text describing Table 3 has been rephrased to (page 14, lines 345-346, new text in bold):

‘For Poland, surplus mortality in 2021 was estimated by the WHO at 100,100 (mean estimate), ranging from 94,500 (low estimate) to 105,700 (high estimate), which is slightly above the number of all COVID-19 mentions registered in that year.’

9. Discussion: in the third paragraph you are refering to Polands excess deaths in 2020, whereas you are not looking at 2020 data beforehand, maybe stick with 2021 here as well? also I was not quite sure where those 67,000 excess deaths come from as they are not displayed in Tab. 3 (though it might be Ref. 50)

Response: Yes, sorry for not making this clear in the original manuscript. The quoted 67 thous. excess deaths in 2020 come from the reference no. 50 (a report by the Ministry of Health of Poland) and to stress this, we modified the manuscript in the following way (pages 16-17, lines 423-427, new text in bold):

‘In these countries, the elevated mortality in the pandemic period should be attributed to other causes of death or to unrecognised COVID-19 cases, but the different diagnostic testing policies implemented by Lithuania and Poland make it difficult to draw broad comparisons49. Nevertheless, a report from the Ministry of Health of Poland50 suggests that the number of unrecognised COVID-19 cases in that country was considerable in 2020, the year for which data on multiple causes of death are not available. According to this analysis, 67,000 excess deaths were reported in 2020, of which 43% were due to COVID-19 as the UCoD and 27% due to other UCoDs in coronavirus patients.’

Note that the Ministry of Health’s estimate of 67,000 excess deaths is similar to the WHO mean estimates shown in Table 3 (63,000), which we believe lends credibility to the Ministry's estimation method.

10. I personally, would have expected at least a little elaborating on other sources of excess mortality (delayed health care access etc.) and even reductions in mortality (e.g. less traffic accidents) in the discussion, you briefly mention those in the beginning but never get back to this and it should at least be acknowledged that other things have happend during pandemic times and not all excess can or should (?) be diretly attributed to COVID-19

Response: Such an acknowledgement was easily incorporated at the end of the discussion; we modified the manuscript as follows (pages 18-19, lines 464-479, new text in bold):

‘During the COVID-19 pandemics, mortality increased far beyond the reported deaths from cor

---

## [Decision Letter · Decision Letter 1]

2 Jul 2024

Unrecognised COVID-19 Deaths in Central Europe: the Importance of Cause-of-Death Certification for the COVID-19 Burden Assessment

PONE-D-24-12758R1

Dear Dr. Fihel,

We’re pleased to inform you that your manuscript has been judged scientifically suitable for publication and will be formally accepted for publication once it meets all outstanding technical requirements.

Kind regards,

Dickens Otieno Onyango

Academic Editor

PLOS ONE

Additional Editor Comments (optional):

Reviewers' comments:

Reviewer's Responses to Questions

**Comments to the Author**

1. If the authors have adequately addressed your comments raised in a previous round of review and you feel that this manuscript is now acceptable for publication, you may indicate that here to bypass the “Comments to the Author” section, enter your conflict of interest statement in the “Confidential to Editor” section, and submit your "Accept" recommendation.

Reviewer #1: All comments have been addressed

Reviewer #2: All comments have been addressed

2. Is the manuscript technically sound, and do the data support the conclusions?

Reviewer #1: Yes

Reviewer #2: Yes

3. Has the statistical analysis been performed appropriately and rigorously? 

Reviewer #1: Yes

Reviewer #2: Yes

4. Have the authors made all data underlying the findings in their manuscript fully available?

Reviewer #1: Yes

Reviewer #2: Yes

5. Is the manuscript presented in an intelligible fashion and written in standard English?

Reviewer #1: Yes

Reviewer #2: Yes

6. Review Comments to the Author

Reviewer #1: no additional comments.................................................................................................

Reviewer #2: This manuscript identifies unrecognised COVID-19 deaths, providing the evidence needed to manage similar epidemics in future, and highlights the limitations of current practises in MCCOD. From this manuscript, we realise that major risk factors for mortality, include cardiovascular disease, neoplasms and obesity, and provides the basis for appropriate public health interventions.

7. PLOS authors have the option to publish the peer review history of their article (what does this mean?). If published, this will include your full peer review and any attached files.

Reviewer #1: No

Reviewer #2: **Yes: **Edwin Walong

---

## [Editor Report · Acceptance letter]

4 Jul 2024

PONE-D-24-12758R1 

PLOS ONE

Dear Dr. Fihel, 

I'm pleased to inform you that your manuscript has been deemed suitable for publication in PLOS ONE. Congratulations! Your manuscript is now being handed over to our production team.

Kind regards, 

on behalf of

Dr. Dickens Otieno Onyango 

Academic Editor

PLOS ONE